# Effects of Parental Workplace Discrimination on Sickness Presenteeism

Joachim Gerich * and Martina Beham-Rabanser

Department of Sociology, Johannes Kepler University, 4040 Linz, Austria; martina.beham-rabanser@jku.at
* Correspondence: joachim.gerich@jku.at

**Abstract:** This paper analyzes the association between experienced and observed parental workplace discrimination and sickness presenteeism. Following stress theoretical approaches and reactance theory, we expected that both experienced and observed parental discrimination of others at the workplace would lead to a reactance behavior and could increase sickness presenteeism, especially in those individuals who deny arguments of justification. Based on survey data from employees aged between 20 and 45 years ($n = 347$), we confirmed experienced discrimination as a double risk factor that goes along with increased sickness, as well as an increased sickness presence propensity. Although observed discrimination against others was unrelated to sickness, it was similarly associated with increased presenteeism. For respondents with their own children, the association between experienced discrimination and presenteeism was amplified in those who disagree with economic justifications of discrimination. The relationship between presenteeism and observed discrimination in childless respondents was amplified in those who appraise discrimination as unfair. In accordance with a stress theoretical approach, we confirm negative health effects of parental discrimination. In accordance with reactance theory, it is concluded that discrimination encourages workers' presenteeism in the sense of a self-endangering behavior to counter inappropriate stereotypes held against them.

**Keywords:** reactance; self-endangering behavior; symbolic amplification; symbolic vilification; presenteeism-propensity



## 1. Introduction

Austrian law (Federal Equal Treatment Act, Federal Law Gazette I No. 66/2004 as amended by Federal Law Gazette I No. 107/2013; Wochenalt and McGrew-Taferl 2020) defines parental workplace discrimination as unequal treatment of employees regarding hiring, remuneration, social benefits, access to vocational training and qualification, promotion and career advancement, termination/dismissal, and other working conditions (such as downgrading, bullying, and allocation of resources and work-related equipment) based on parental status and caregiving responsibilities. This also includes unequal treatment based on inferred prospective caring responsibilities of women in their reproductive years. According to Henle et al. (2020, 59f), parental workplace discrimination covers intentional unequal treatment of caregivers "due to stereotypes about how they will perform in the workplace because of their caregiving role" and organizational cultures and policies "that put caregivers at a disadvantage". Moreover, Austrian mothers and fathers with caring responsibilities for children have a statutory right to request reduced working hours under certain conditions. According to the Maternity Protection Act 1979, employees in companies with more than 20 employees are allowed to reduce their working hours after the birth of their child until the child's seventh birthday if they have been continuously employed in the same company in the last three years. In doing so, parents must reduce their working hours by at least 20 percent. The start, duration, and extent of the part-time employment must be arranged with the employer, taking into account the interests of the company as well as the employee's interests (§ 15h Maternity Protection Act 1979, Federal Law Gazette

221/1979 amended by Federal Law Gazette I 149/2015). Therefore, in many companies, denying the request of parents with preschool children to reduce working hours contradicts the legal provisions in Austria and represents a form of parental discrimination.

The number of publications on parental workplace discrimination has increased in recent years (Becker et al. 2019; Hipp 2020; Shen and Dhanani 2018; Wochenalt and McGrew-Taferl 2020). However, there is a distinct lack of studies on possible effects of these unequal treatments on health and health-related behavior such as sickness presenteeism, defined as working while ill (Ruhle et al. 2020). While some researchers have emphasized the positive effects of presenteeism, such as maintenance of social contacts, fulfillment of psychological needs, and the perpetuation of career opportunities despite sickness (Biron et al. 2022; Karanika-Murray and Biron 2020; Lohaus et al. 2020), other studies have found that presenteeism is related to detrimental health effects (such as mental and physical health problems, cardiovascular diseases, or burnout) in the long run (Bergström et al. 2009; Demerouti et al. 2009; Kivimaki et al. 2005; Skagen and Collins 2016; Taloyan et al. 2012). Thus, presenteeism must be regarded as a health-related risk behavior.

The research presented in this paper aims to analyze the association between experienced and observed parental workplace discrimination and sickness presence behavior of employees. This research should help to uncover pathways that motivate workers to attend work despite sickness, in light of experienced or observed parental workplace discrimination. Regarding such pathways, we develop and test a theoretical model that combines stress-theoretical explanations with assumptions following reactance and injustice theory.

### 1.1. Effects of Discrimination on Health and Sickness Presenteeism: State of Research and Theoretical Explanations

Meta-analytic research on discrimination related to race and sex (Jones et al. 2016; Lee and Ahn 2011) confirmed evidence for the negative effects of overt as well as subtle discrimination on physical and mental health. With respect to presenteeism, two related but different concepts must be distinguished (Ruhle et al. 2020). One strand of research defines presenteeism as productivity loss due to health problems. The other strand of research is focused on the act of sickness presenteeism (that is, going to work while ill), which is also the focus of the present paper. Concerning presenteeism defined as productivity loss due to health problems, previous research (Cho et al. 2016; Yang et al. 2016) has established a positive association between experienced discrimination based on age, education, religion, and sexuality, on one hand, and presenteeism, on the other. Similarly, Deng et al. (2020) found that global perceptions of everyday discrimination were directly and indirectly (through increased negative and decreased positive affect) related to increased presenteeism. Furthermore, a positive relationship between discrimination and presenteeism with respect to the act of sickness presenteeism was confirmed in meta-analytic research (Miraglia and Johns 2016).

Typically, stress-theoretical explanations are proposed regarding possible mechanisms linking experiences of discrimination with presenteeism behavior. Miraglia and Johns (2016) proposed a dual-path framework for explaining presenteeism behavior. Within this framework, they assumed two distinct mechanisms as drivers for sickness presenteeism. On one hand, the extent of sickness presenteeism depends on an individual's health state. Hence, factors that negatively affect individuals' health will increase sickness presence (but also sickness absence) through the health impairment path. On the other hand, sickness presence is thought to be promoted through a motivational path, where highly committed and engaged workers attend work despite illness based on felt obligations and intrinsic motivation to spend extra energy on work. Hence, work-related factors that increase intrinsic work motivation and positive affect are thought to increase sickness presence (but probably decrease sickness absence) through the motivational path. Based on this dual-path framework, Miraglia and Johns (2016) argued that discrimination and harassment—similar to hindrance job demands—will damage motivation and commitment. Hence, a negative rather than a positive effect on presenteeism through the motivational path is expected.

However, as discrimination can provoke stress and impaired health, the positive effects of discrimination on presenteeism (as well as on sickness absence) through the health impairment path are expected. Based on meta-analytic data, Miraglia and Johns (2016) confirmed a moderate positive effect of discrimination on sickness presenteeism. Similarly, Yang et al. (2016) argued that discrimination is a stress-related work-factor, which increases presenteeism due to health impairments. However, in research on workplace sexism, Manuel et al. (2017) also confirmed that sexism experience is related to increased sickness presenteeism mediated by poor health, although they found sexism to be unrelated to sickness absence. Hence, other additional mechanisms beside the stress-induced health-impairment path must be considered to explain the shift from sickness absence to sickness presence associated with increased illness as a consequence of experienced discrimination.

In an experimental study on parental discrimination, Heiserman and Simpson (2022) told participants with caring responsibilities who applied for a job (offered as either higher or lower status positions) that they were assigned to tasks corresponding to a lower status position due to their parental status, because it is assumed that parents will be unable to show enough engagement necessary for the higher job position. As expected, those who received the stigmatizing message rated their jobs worse, reported less interest, and perceived a lower status position than a control group who received a neutral message reasoning their assignment to the lower status group. However, contrary to their expectations, higher job performance was found among participants who received the stigmatizing message compared to the neutral message. Reactance theory (Brehm 1966; Kray et al. 2001) may be employed to explain this result, where individuals who perceive that their ability and motivation is questioned increase their efforts to disprove inappropriate stereotypes against them. Thus, increased sickness presenteeism associated with experienced (parental) discrimination may be expected to be a type of self-endangering behavior (Dettmers et al. 2016). Self-endangering behavior in such a form can be defined as an individual's intentional problem-focused coping strategy, where the maintenance of productivity and organizational reputation is prioritized over personal health in order to disprove stereotypes regarding low performance and unreliability against them.

Dhanani et al. (2018) focused on the health effects of workplace discrimination. In addition to stress-theoretical mechanisms, they suggested that the health-related effects of discrimination were caused by perceived injustice that violates norms of reciprocity. A large body of earlier research has shown evidence for the detrimental health effects of violated work-related reciprocity (Robbins et al. 2012; Siegrist 2010; van Vegchel et al. 2005). Apart from negative health effects induced by such perceived violations of reciprocity, it may also trigger reactance behavior such as that described above. This can be assumed because reactance behavior to parental discrimination, as described above, can only be expected when individuals appraise incidents of discrimination as inappropriate and unfair. Conversely, when individuals think that parental discrimination is justified, there will be no need to counter inappropriate stereotypes.

### 1.2. Appraisal of Parental Workplace Discrimination

As other scholars have already stated, individuals' perceptions of unfairness and illegitimacy of unequal treatment and labeling discrimination as such are, for instance, prerequisites of the willingness for seeking legal redress in those affected by discrimination (Heiserman and Simpson 2022; Hirsh and Lyons 2010; O'Connor and Kmec 2020). Consequently, we expect that elevated sickness presenteeism as reactance behavior to parental workplace discrimination is only observed in those individuals who judge the unequal treatment as unfair and illegitimate. However, there is reason to assume that, beside evaluations regarding fairness and legitimacy, other evaluative considerations may dampen or amplify individuals' reactance when faced by incidents of discrimination.

In a qualitative research of parental discrimination claims, Byron and Roscigno (2014) found that employers frequently adopt two strategies to justify unequal treatment. With the first strategy (symbolic vilification), it is argued that mothers or pregnant employees

are basically less committed to work, less productive, and less reliable due to their increased focus on their offspring. Such arguments build on stereotypes, whereupon parents are less productive and motivated in a "natural" sense. With the second strategy (symbolic amplification), unequal treatment is justified on the basis of economic principles, where parenthood of employees is qualified as an economic liability, and therefore parental discrimination is presented as a rational decision following a business logic that must prioritize economic efficiency over worker benefits or reconciliation of work and family. Byron and Roscigno (2014) argued that justifications following symbolic vilification and amplification are partly successful because they are consistent with widespread accepted societal values—for instance, traditional gender-role expectations or favorable attitudes toward marketization and meritocratic ideals. Therefore, we expect that some individuals will show resonance with justifying arguments following symbolic vilification and amplification. Moreover, we suppose that sickness presenteeism is a reactance behavior that is less likely to be observed in those individuals who show resonance with these justifying arguments, and more likely in those who deny such arguments.

*1.3. Perceived versus Observed Discrimination*

Although previous discrimination research primarily focused on outcomes of direct personal experienced discrimination, some findings give reason to assume similar effects induced by observed discrimination of others. Based on a systematic literature review and their own experimental studies, Ozier et al. (2019) confirmed similar detrimental effects of experienced and observed racial discrimination on executive cognitive functions necessary for focusing, self-control, decision making, and problem-solving. Moreover, as a consequence of impaired cognitive functioning, the authors expected negative long-term effects on health, educational attainment, and employment as consequences of experienced and observed discrimination. Good et al. (2012) analyzed mechanisms relevant for confronting perpetrators following experienced and observed sexism. They concluded similar promoting and impeding factors of confrontation for experienced and observed sexism.

Correspondingly, in their research on workplace discrimination, Dhanani et al. (2018) expected negative health effects of experienced workplace discrimination—defined as the perceived direct unequal treatment targeted toward the focal person—as well as of observed discrimination, defined as the general prevalence of discrimination perceived at the workplace. Although the authors assume negative health effects induced by both forms of discrimination, they expected stronger effects from experienced discrimination compared to observed discrimination. Observed mistreatment of others may be also attributed to alternative causes (that is, shortcomings of individuals affected), which reduces the perceived injustice of observed discrimination and, according to the authors, explains the expected discrepancy in effect. Hence, it is argued that experienced discrimination will be appraised as more unfair than observed discrimination and, therefore, a weaker effect on health and health-related behavior of the latter is expected. Whereas (as expected) their meta-analytic results confirmed a stronger relationship of experienced compared to observed discrimination with job stress, the opposite (that is, a stronger relationship with observed compared to experienced discrimination) was found with respect to turnover intentions and mental health.

Consequently, given our assumption outlined above whereupon individuals faced by parental workplace discrimination may increase sickness presenteeism as reactance behavior, we have reason to expect that such self-endangering overcommitment may similarly be induced by observed parental workplace discrimination of others at the present occupation.

Although both experienced and observed discrimination may provoke similar effects, it must be considered that both refer to substantially different analytical layers. Whereas experienced discrimination refers to the individual system of lifetime experienced mistreatment, observed discrimination refers to a property of a social setting such as organizational practices and organizational culture. Despite similar outcomes of experienced discrimination (at the individual level) and observed discrimination (at the organizational level), the

mechanisms behind these outcomes may differ. For instance, Ozier et al. (2019, p. 1105) stated that while experienced discrimination makes individuals more vigilant to mistreatment of the self, "those who observe discrimination may also become more vigilant to cues that suggest that members of their group (and/or members of other stigmatized groups) may be discriminated against". Therefore, we can expect that the group of people that may be affected by observed discrimination is not necessarily limited to those with similar attributes as those mistreated (such as those with caring responsibilities, pregnant women, or those who plan to have children). Instead, especially those who appraise observed parental workplace discrimination as inappropriate and unfair may find themselves placed in a hostile and distrusting environment, which may induce self-endangering behavior even when individuals are not part of the targeted group of observed mistreatments.

*1.4. Research Model*

To summarize, it is expected that experienced and observed parental workplace discrimination impairs employees' health, leading to a higher number of days with sickness.

**H1.** *Observed and experienced parental workplace discrimination is positively related to the number of days with sickness.*

Furthermore, according to assumptions of reactance theory, discrimination is expected to trigger behavior that counters stereotypical allegations concerning low motivation, performance, and commitment. Hence, discrimination may influence individuals' decision process between sickness absence and sickness presence. More precisely, it is assumed that perceived discrimination increases presenteeism propensity (Gerich 2014, 2016), which is defined as the likelihood of choosing sickness presence instead of sickness absence in case of a health event. In other words, we expect that the health impairment path induced by experienced or observed parental workplace discrimination is accompanied by a reactive path, which is thought to shift the proportion of total sickness days from sickness absence to sickness presence.

**H2.** *Observed and experienced parental workplace discrimination is positively related to the number of sickness presence days.*

**H3.** *Observed and experienced parental workplace discrimination is positively related to presenteeism propensity.*

Moreover, as the effect on presenteeism propensity is thought to be caused by perceived injustice, it is expected that the effect of perceived discrimination on presenteeism propensity is amplified in individuals who perceive greater injustice related to unequal treatment of workers based on their parental status.

**H4.** *There is an interaction effect between experienced and observed parental workplace discrimination and appraisal of parental workplace discrimination such that the positive association between workplace discrimination and presenteeism propensity is amplified with stronger perceptions of unfairness, illegitimacy, and stronger rejection of symbolic vilification and symbolic amplification.*

## 2. Materials and Methods

The study presented in this paper is based on an analysis of survey data collected from a sample of employees between 20 and 45 years of age drawn from the register of the Upper Austrian Chamber of Labor who were invited by postal mail to join a Web survey about fairness to parents in the working context. The sample was disproportionally stratified by gender (80 percent females and 20 percent males), as women are disproportionately more affected by parental workplace discrimination. Membership with the Chamber of Labor is mandatory for most Austrian employees; hence, the sample consists of a heterogeneous group of employees from different occupations and branches. Analyses were based on

*n* = 347 answered questionnaires. Due to item nonresponse, the number of cases was reduced for some analyses (see Table 1 for details). Among the respondents, 11.2 percent were males and 88.8 percent were females; 49.6 percent reported having children. The mean age of respondents was 32.05 years (SD = 7.04).

**Table 1.** Descriptives.

| | Total Sample | | Respondents with Children | | Childless Respondents | | *p* |
|---|---|---|---|---|---|---|---|
| | Mean/Percent (SD) | n | Mean/Percent (SD) | n | Mean/Percent (SD) | n | |
| Observed discrimination | 1.93 (0.66) | 347 | 1.93 (0.71) | 172 | 1.93 (0.62) | 175 | 0.992 |
| Experienced discrimination | 0.22 (0.20) | 347 | 0.27 (0.20) | 172 | 0.17 (0.20) | 175 | <0.001 |
| Symbolic vilification | 1.22 (0.44) | 344 | 1.19 (0.37) | 170 | 1.24 (0.50) | 174 | 0.350 |
| Symbolic amplification | 1.73 (0.74) | 345 | 1.65 (0.68) | 171 | 1.80 (0.79) | 174 | 0.051 |
| Fairness | 3.60 (0.63) | 344 | 3.54 (0.65) | 171 | 3.65 (0.61) | 173 | 0.110 |
| Legitimacy | 3.64 (0.64) | 345 | 3.59 (0.67) | 170 | 3.69 (0.61) | 175 | 0.170 |
| Sex (male) | 11.2% | 347 | 8.1% | 172 | 14.3% | 175 | 0.070 |
| Age | 32.05 (7.04) | 347 | 36.06 (5.79) | 172 | 28.11 (5.85) | 175 | <0.001 |
| Job satisfaction | 3.56 (1.18) | 347 | 3.43 (1.26) | 172 | 3.70 (1.09) | 175 | 0.035 |
| Educational level | | 347 | | 172 | | 175 | 0.086 |
| obligatory | 2.0% | | 2.3% | | 1.7% | | |
| apprenticeship/vocational school | 37.2% | | 43.6% | | 30.9% | | |
| high school | 28.2% | | 24.4% | | 32.0% | | |
| university | 33.6% | | 29.7% | | 35.4% | | |
| Sickness absence days | 5.84 (7.02) | 333 | 5.01 (6.12) | 164 | 6.65 (7.73) | 169 | 0.032 |
| Sickness presence days | 7.15 (7.91) | 330 | 7.52 (8.75) | 161 | 6.79 (7.03) | 169 | 0.404 |
| Total sickness days | 12.94 (11.75) | 324 | 12.54 (12.45) | 161 | 13.32 (11.07) | 166 | 0.552 |
| Presenteeism propensity | 0.52 (0.34) | 297 | 0.56 (0.34) | 142 | 0.49 (0.33) | 155 | 0.071 |

*p*: Two-sided significance of group differences (*t*-test for mean-differences and $\chi^2$-test for categorial variables).

### 2.1. Measures

### 2.1.1. Dependent Variables

*Sickness absence* was measured with the question "Approximately how many days did you take as sick leave during the past 12 months?"

The question regarding *sickness presenteeism* was "Approximately how many days did you attend work during the past 12 months, even though your health state would have justified taking sick leave?"

To avoid bias due to outliers and cases with long-term sickness, responses exceeding 60 days on each of the both questions (sickness absence and sickness presence) were excluded (Gerich 2016; Lohaus et al. 2020).

The total *number of days with sickness* was estimated as the sum of sickness absence and sickness presence days.

Moreover, the key question in the present research is whether experienced or observed discrimination affects individuals' decision between sickness absence and sickness presence in times of sickness. For that purpose, *presenteeism propensity* is calculated as the number of sickness presence days divided by the sum of sickness absence and sickness presence days (Gerich 2014; Ruhle et al. 2020). Contrary to the frequency measures (sickness absence and sickness presence days), presenteeism propensity focuses on the decision process between absence and presence when individuals are faced by sickness and—different to sickness presence days—this measure is not confounded by the extent of individuals' sickness. The value range of presenteeism propensity is between 0 (all sickness days are spent as sick leave) and 1 (all sickness days are spent in presence). For example, the sample mean propensity of 0.52 found in our data means that the average probability of deciding for sickness presence instead of sickness absence is estimated as 52 percent. Presenteeism propensity is only computable for those respondents whose total number of days with sickness is larger than zero, although this is reasonable given that valid information on the decision process is only available for those who experienced sickness during the past year.

### 2.1.2. Independent Variables

*Experienced parental workplace discrimination* was measured using 49 items that scoped respondents' experience with unequal treatment related to parental status. Items were grouped by different modalities and biographic stations (unequal treatment concerning job application and job interviews, rejection of an application, termination, parental part-time, return from parental leave, and everyday discrimination due to parenthood). The items were lent toward different types of parental discrimination as listed in the Austrian Federal Treatment Act and a documentation of frequent cases from an advocacy organization (Wochenalt and McGrew-Taferl 2020) and were developed by the authors of this paper together with two experts in the field with professional experience in counselling victims of parental discrimination. For example, items asked about downgrading after return from parental leave, dismissals due to parenthood or due to requests for part-time employment or parental leave, mobbing from colleagues or supervisors due to caring responsibilities, or whether questions regarding care responsibilities, pregnancy, or family planning were asked during job interviews.

For logical reasons, some items only applied to male or female respondents and some modalities only applied to those with caring responsibilities for children aged 15 years or younger (labelled as respondents with children thereafter). However, childless respondents were able to answer items concerning job application, rejection of an application, and termination. Nevertheless, because different mechanisms may be relevant for people with and without children, separate analyses were performed for both subsamples. Depending on the modalities, different answer categories needed to be applied. For instance, answer categories for items regarding questions asked during job interviews were (1) such questions were never asked, (2) such questions were asked indirectly, and (3) such questions were asked directly. On the contrary, answer categories for other modalities (such as downgrading after return from parental leave) ranged from (1) not at all to (4) fully applies. For the computation of an overall index of experienced discrimination, all answer categories were transformed to continuous values between 0 (representing no experience) and 1 (highest intensity of experienced discrimination) (i.e., 3-point scales: original values 1, 2, 3 were transformed to 0, 0.5, 1; 4-point scales: original values 1, 2, 3, 4 were transformed to 0, 0.33, 0.66, 1). Subsequently, the index was computed as the mean score across all items (Cronbach's Alpha = 0.93).

Sixteen items were used to assess observed parental workplace discrimination at respondents' current workplace. Sample items are "workers without caring responsibilities are preferred for promotion or leader positions" and "male colleagues who request parental leave or a reduction in working hours will likely be dismissed". Answer categories ranged from (1) not at all to (4) fully applies. Observed parental workplace discrimination was computed as the mean response on the 16 items (Cronbach's Alpha = 0.91).

### 2.1.3. Moderators

According to Austrian laws, women in their reproductive years must not be discriminated based on their prospective caring responsibilities (Equal Treatment Act, Federal Law Gazette I No. 66/2004 as amended by Federal Law Gazette I No. 107/2013). In addition, mothers and fathers with preschool children have a statutory right to request a reduction in their working hours under certain conditions to make it easier for parents to reconcile their work and family duties (§ 15h Maternity Protection Act 1979, Federal Law Gazette 221/1979 amended by Federal Law Gazette I 149/2015). Based on these legal regulations, we used two fictional cases to measure respondents' appraisal of unequal treatment. Although both cases represent an illegal treatment due to parental status according to existing laws, we expect variations of respondents' perceptions of whether such treatment is fair or acceptable.

While case study research is an increasingly popular approach among qualitative researchers (Creswell 2013; Priya 2021; Thomas 2011), case study approaches in quantitative studies are still rare. However, fictional case studies seem to be a suitable approach to evaluate tendencies towards stereotypical judgments, which are methodologically difficult to capture in real settings (Gorman and Mosseri 2019; Kaiser and Major 2006; O'Connor and Kmec 2020). Therefore, we developed our measures on previous qualitative studies (e.g., Byron and Roscigno 2014).

For the measurement of respondents' *appraisal of unequal treatment* due to parental status, we constructed two fictional cases. Case 1 details a situation where a job application of a young, qualified female was rejected, as the employer wanted to avoid the potential complication of an unexpected employee pregnancy. Case 2 features a male employee with caring responsibilities whose request for reduced working hours was rejected. We asked respondents to rate both fictional cases with respect to symbolic vilification ("The decision is comprehensible, because <mothers>/<this man> show(s) little interest in work"), symbolic amplification ("The decision is understandable, because it causes economical disadvantages for the enterprise"), fairness ("The decision is unfair"), and legitimacy of the treatment ("I feel that such a procedure should be forbidden by law" for Case 1 and "I feel the father should have a legal right for a part-time employment" for Case 2). All items were followed by a four-point Likert-type response-scale ranging from (1) "I do not agree at all" to (4) "I fully agree". Mean ratings from both cases for each dimension were computed.

### 2.1.4. Control Variables

Beside respondents' sex and age, respondents' job satisfaction and educational status were considered as control variables.

*Job satisfaction* was considered as a variable to control for possible effects of workplace discrimination on presenteeism that could be explained through a motivational path (Miraglia and Johns 2016). Job satisfaction was measured with a single item ("Taking into consideration all things: How satisfied are you with your job?) on a five-point rating scale ranging from (1) "very unsatisfied" to (5) "very satisfied" (Quinn et al. 1974). Other research (Fakunmoju 2020; Wanous et al. 1997) has indicated that single-item measures of job satisfaction performed equally well as multiple item measures.

*Educational status* (categorized as obligatory, apprenticeship/vocational school, high school, university) was considered a control variable because previous research found higher sickness absence and sickness presence rates (Aronsson et al. 2000; Hansen and Andersen 2008; Johansson and Lundberg 2004; Löve et al. 2013; Seglem et al. 2020), as well

as higher risks of reported discrimination in lower socioeconomic status groups (Potter et al. 2019; Stepanikova and Oates 2017).

*2.2. Analyses*

In investigating hypotheses H1 to H3 as outlined above, associations between experienced and observed parental discrimination and the number of days with sickness, sickness absence, sickness presence, and presenteeism propensity are evaluated with help of bivariate correlations and multivariate regression analyses (OLS). In investigating H4, interaction effects between experienced and observed parental discrimination and the four considered dimensions of appraisal of unequal treatment with respect to presenteeism propensity are examined. As differences between childless respondents and those with children were expected, separate analyses with respect to these groups were conducted. Because the distributional requirements to compute Pearson's correlation and OLS-regressions may be violated, robustness checks by using alternative model-estimations were conducted. With respect to the count measures (days with sickness, sickness absence, sickness presence), the analyses were replicated with zero-inflated negative binomial regression models (Bierla et al. 2013; Gerich 2016). Presenteeism propensity is a continuous proportional measure including one and zero cases. Therefore, a generalized linear model with a binomial probability distribution and a logit link function was used for the robustness-check regarding for this outcome (Papke and Wooldridge 1996). All bivariate correlations were replicated by Spearman's Rho coefficient. The results of these robustness-checks are available from the supplementary material (Tables S1 and S2). As the results of the robustness checks revealed no substantial differences compared to the OLS regressions, we use the results of the latter due to better interpretation. The main analyses were performed with SPSS 29 (IBM Corp. 2022), and the robustness-checks were performed with Stata SE 14 (Stata Corp. 2015).

## 3. Results

Descriptive information for the total sample, as well as for the subgroups of childless respondents and those with own children, can be seen in Table 1.

As expected, childless respondents reported less experienced discrimination than those with their own children. Compared to respondents with own children, childless respondents were younger in age, and reported higher job satisfaction and a higher number of sickness absence days.

Table 2 shows bivariate and multivariate analyses for the total sample concerning H1 to H3.

According to these results, higher levels of experienced and observed parental workplace discrimination are related to a higher number of total days with sickness. However, only the relationship with experienced discrimination is confirmed significant after adjusting for the control variables. Therefore, hypothesis H1 is partially confirmed. Whereas observed and experienced discrimination are not significantly related to the number of sickness absence days, positive association between both forms of discrimination and sickness presence days are confirmed. Hence, the additional days with sickness associated with parental workplace discrimination are mainly spent in sickness presence, but not in absence. This is reflected in a positive association between observed and experienced parental workplace discrimination and presenteeism propensity. Hence, in accordance with H2 and H3, higher experienced and stronger observed discrimination are associated with a higher tendency to decide for sickness presence instead of sickness absence when individuals are faced with sickness.

Separate analyses for those with and without children are shown in Table 3.

The multivariate results for the group of childless respondents show that only experienced (but not observed) workplace discrimination is significantly related to a higher number of days with sickness, a higher number of presenteeism days, and a higher presenteeism propensity.

The results for the group of respondents with own children are somewhat mixed. On one hand, similar to the group of childless respondents, only significant relationships between experienced discrimination and the number of days with sickness and the number of sickness presence days are confirmed. However, only experienced discrimination in the unadjusted model and only observed discrimination in the adjusted model are significantly related to presenteeism propensity, although the smaller sample size of the separated subgroup analyses must be considered.

**Table 2.** Bivariate and multivariate results (total sample).

| | Presenteeism Propensity | | | Sickness Presence Days | | | Sickness Absence Days | | | Sickness Days | | |
|---|---|---|---|---|---|---|---|---|---|---|---|---|
| | I | II | III | I | II | III | I | II | III | I | II | III |
| Observed discrimination | 0.24 *** | 0.14 * | 0.16 * | 0.28 *** | 0.18 *** | 0.15 * | 0.06 | 0.07 | 0.00 | 0.23 *** | 0.16 ** | 0.10 |
| Experienced discrimination | 0.30 *** | 0.24 *** | 0.19 ** | 0.33 *** | 0.26 *** | 0.26 *** | 0.01 | −0.01 | 0.03 | 0.23 *** | 0.17 ** | 0.19 ** |
| Sex (male) | | | −0.12 * | | | 0.04 | | | 0.10 | | | 0.09 |
| Age | | | 0.02 | | | −0.05 | | | −0.15 ** | | | −0.14 * |
| Job satisfaction | | | 0.02 | | | −0.07 | | | −0.08 | | | −0.08 |
| Educational level | | | | | | | | | | | | |
| obligatory | | | −0.02 | | | −0.01 | | | 0.15 ** | | | 0.08 |
| apprenticeship/vocational school | | | 0.10 | | | 0.15 * | | | 0.07 | | | 0.13 * |
| high school | | | 0.04 | | | 0.01 | | | −0.02 | | | −0.01 |
| university | | | ref. | | | ref. | | | ref. | | | ref. |
| R² | | 0.11 | 0.12 | | 0.14 | 0.17 | | 0.00 | 0.07 | | 0.08 | 0.14 |
| n | | 297 | | | 330 | | | 333 | | | 324 | |

I: bivariate correlations; II: multivariate regression (OLS, standardized coefficients); III: multivariate regression (OLS, standardized coefficients, adjusted for controls); * $p: \leq 0.05$; ** $p: \leq 0.01$; *** $p: \leq 0.001$.

**Table 3.** Bivariate and multivariate results separated for childless respondents and respondents with children.

| | Presenteeism Propensity | | | Sickness Presence Days | | | Sickness Absence Days | | | Sickness Days | | |
|---|---|---|---|---|---|---|---|---|---|---|---|---|
| | I | II | III | I | II | III | I | II | III | I | II | III |
| *with children* | | | | | | | | | | | | |
| Observed discrimination | 0.28 *** | 0.18 | 0.21 * | 0.32 *** | 0.19 * | 0.16 | 0.09 | 0.04 | 0.00 | 0.29 *** | 0.17 | 0.13 |
| Experienced discrimination | 0.29 *** | 0.20 * | 0.15 | 0.36 *** | 0.26 ** | 0.28 ** | 0.11 | 0.09 | 0.11 | 0.32 *** | 0.24 ** | 0.26 ** |
| R² | | 0.11 | 0.14 | | 0.15 | 0.20 | | 0.01 | 0.05 | | 0.12 | 0.18 |
| n | | 142 | | | 161 | | | 164 | | | 158 | |
| *childless* | | | | | | | | | | | | |
| Observed discrimination | 0.19 ** | 0.12 | 0.12 | 0.22 ** | 0.15 | 0.11 | 0.05 | 0.05 | −0.01 | 0.16 * | 0.12 | 0.05 |
| Experienced discrimination | 0.27 *** | 0.24 ** | 0.20 * | 0.30 *** | 0.26 *** | 0.24 ** | −0.01 | −0.03 | 0.04 | 0.17 * | 0.14 | 0.17 * |
| R² | | 0.09 | 0.13 | | | | | 0.00 | 0.16 | | 0.04 | 0.19 |
| n | | 155 | | | 169 | | | 169 | | | 166 | |

I: bivariate correlations; II: multivariate regression (OLS, standardized coefficients); III: multivariate regression (OLS, standardized coefficients, adjusted for controls); * $p: \leq 0.05$; ** $p: \leq 0.01$; *** $p: \leq 0.001$.

Concerning H4, interaction terms between the four considered dimension of discrimination appraisal (that is, illegitimacy and fairness perceptions, as well as agreement with symbolic vilification and symbolic amplification) and experienced and observed discrimination have been added to Model III predicting presenteeism propensity. None of the assumed interaction effects were confirmed as significant for the total sample.

However, two of the proposed interaction effects were confirmed in the separate analyses of both subgroups, which are shown in Figures 1 and 2.

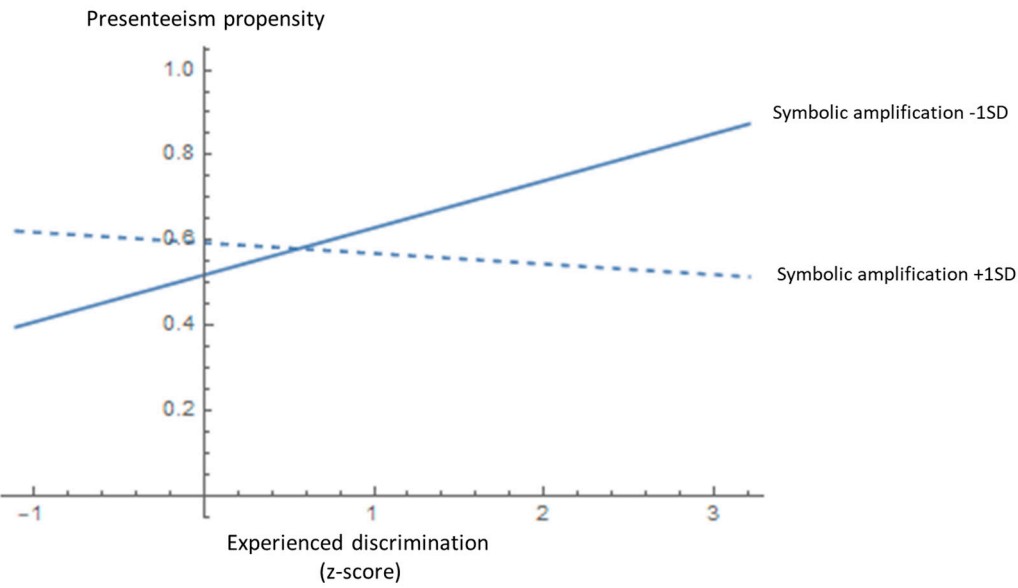

**Figure 1.** Interaction effect between symbolic amplification and experienced discrimination (respondents with children).

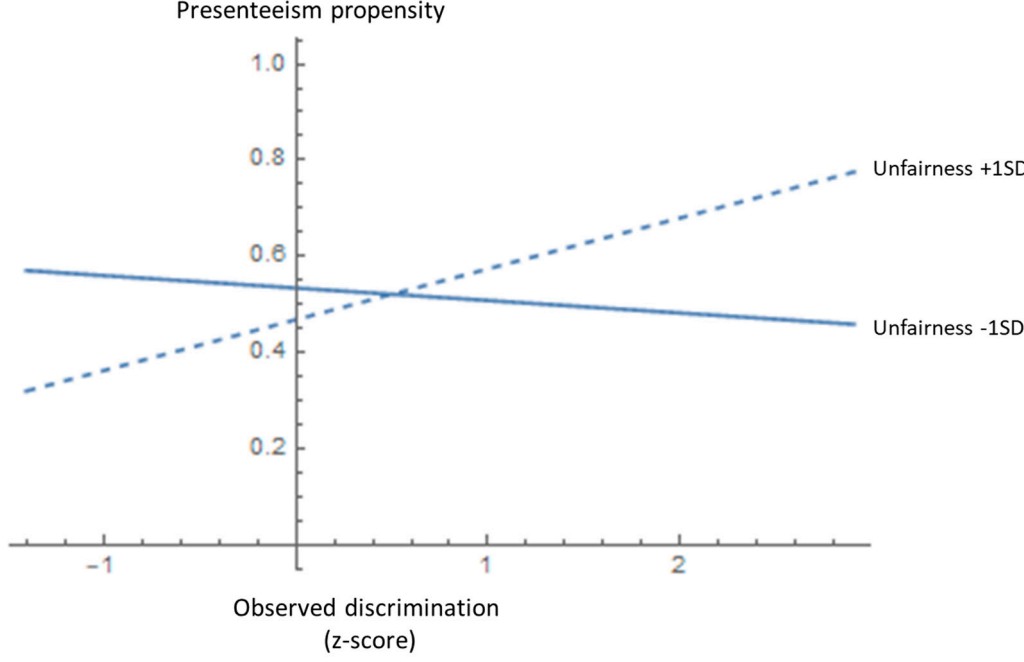

**Figure 2.** Interaction effect between judgments of unfairness and observed discrimination (childless respondents).

For the subsample of respondents with children, a significant interaction effect between experienced discrimination and their resonance with symbolic amplification on presenteeism propensity was confirmed (OLS: β = −0.24; *p* = 0.017; Binomial regression: B = −0.32; *p* = 0.003). For the subsample of childless respondents, a significant interaction effect of observed discrimination and the judgment of unfairness (OLS: β = 0.22; *p* = 0.010; Binomial regression: B = 0.28; *p* = 0.005) was confirmed.

Figure 1 shows an increasing presenteeism propensity with higher levels of experienced discrimination in employees with children who disagree with arguments of symbolic amplification. On the contrary, experienced discrimination is unrelated to presenteeism when individuals show resonance for symbolic amplification.

Figure 2 shows that observed discrimination is associated with higher presenteeism propensity in employees without children, when unequal treatment in the hypothetical scenarios is judged as unfair, but not when it is judged as fair.

To summarize the results, experienced and observed parental workplace discrimination are both associated with an elevated presenteeism propensity. However, in the group of people with their own children, the association with experienced discrimination is contingent on their disagreement with justifications based on economic considerations (symbolic amplification). In the group of childless respondents, the association with observed discrimination is contingent on their unfairness appraisal of parental discrimination.

## 4. Discussion

The research presented in this paper aimed to analyze possible associations between experienced and observed parental workplace discrimination and increased sickness presenteeism behavior.

We found that parental workplace discrimination experienced during an occupational biography is related to impaired health (indicated by an increased number of days with sickness) and, consequently, with an elevated number of sickness presence days. However, the increased number of sickness days was not mirrored in the number of sickness absence days, indicating a substitution of sickness absence by sickness presence. In accordance with this finding, we confirmed a significant association between experienced parental workplace discrimination and increased presenteeism propensity, which is a measure estimating individuals' tendency to decide for sickness presence instead of sickness absence on days with health complaints. Thus, we found evidence that experienced parental workplace discrimination acts as a "double risk factor" (Aronsson and Gustafsson 2005) that increases the risk of illness and, at the same time, increases a shift from sickness absence to sickness presence behavior. As already noted by Hansen and Andersen (2008, p. 958), double risk factors are related to sickness presenteeism "by more than one causal mechanism". We argue that a stress-theoretical explanation is a suitable approach in explaining the health-related pathway (that is, increased sickness presenteeism as a consequence of impaired health). However, stress theoretical approaches are not sufficient to explain a shift from sickness absence to sickness presence. Instead, we assumed that sickness presenteeism is also driven by a reactance behavior to inappropriate stereotypes held against an individual. In such a perspective, sickness presence is explained as a self-endangering behavior where individuals intentionally prioritize the maintenance of their goal attainment, career prospective, and organizational reputation over personal health to disprove stereotypes regarding low reliability and performance held against them. Such an explanation also aligns with findings from other experimental research (Heiserman and Simpson 2022), showing that parental workplace discrimination impairs workers job satisfaction and motivation but, at the same time, increases performance.

Contrary to experienced discrimination, observed parental discrimination (that is, incidents of discrimination of others observed at the current workplace) was not related to the number of days with sickness based on multivariate results (neither for the total sample nor within both subgroups analyzed). However, similar to experienced discrimination, observed parental discrimination was significantly related to increased sickness presenteeism propensity over and above exposure to experienced discrimination (based on the analyses for the total sample and within the subsample of those with children). Hence, although observed discrimination seems not to be effective through a health impairment path, it still seems to trigger sickness presenteeism. This is consistent with findings and suggestions from other research, according to which observed and experienced discrimination share some common outcomes, albeit due to probably different mechanisms involved (Dhanani et al. 2018; Good et al. 2012; Ozier et al. 2019). With respect to our findings, we suggest that observed discrimination may not be seen as direct impeachment of the self and therefore not effective as a relevant health-affecting stressor. Nevertheless, observed discrimination may be seen as an organizational culture, where esteem and career opportunities are not

evaluated from workers' actual commitment and productivity, but are inferred from being a member of a specific group (that is, those with caring responsibilities). Consequently, and as with experienced discrimination, such organizational cues may induce overcommitment in order to signalize engagement.

Moreover, building on the theoretical perspective of the reactance framework, we assumed that sickness presenteeism as reactive behavior follows parental discrimination only in individuals who evaluate such mistreatment as inappropriate. This was expected because, from a theoretical view, stereotypes need to be perceived as inappropriate in order for them to induce reactance behavior. For this purpose, we asked respondents to evaluate two fictional cases of parental discrimination. Both cases describe illegal parental treatment according to existing laws in Austria. Respondents evaluated the cases with respect to fairness and legitimacy, as well as their alignment with justifying arguments following symbolic vilification (that is, justifying discrimination by lower work commitment of those with caring responsibilities) and symbolic amplification (that is, justifying discrimination as a rational decision according to economic needs and business logics). Indications in that direction were confirmed by a significant interaction effect found in the subgroup of respondents with their own children. These results show that experienced parental discrimination is not related to an increased presenteeism propensity in those parents who accept justifying arguments of symbolic amplification. On the contrary, experienced parental discrimination is related to an elevated presenteeism propensity in those who deny such attempts of justification. We did not expect that the interaction effect would be exclusively limited to the appraisal dimension of symbolic amplification. Of course, methodological reasons—such as low statistical power in testing interaction effects (Aiken and West 1991)—may explain why interactions with respect to the other three dimensions (fairness, legitimacy, and symbolic vilification) were not confirmed. An alternative explanation could be that unlike the other dimensions, symbolic amplification is an attempt to justify unequal treatment that is argued by external needs (business principles) that are not necessarily perceived to be related to the self. Hence, those who agree with such justifications may feel disadvantaged but not mistreated due to an accepted priority of "higher principles". Consequently, reactance behavior following experienced parental workplace discrimination may be dampened in individuals who strongly align with values of meritocratic orders and principles of market logics, but amplified in those who reject such values.

Similarly, an interaction effect in the group of childless respondents was confirmed such that observed discrimination is related to an increased presenteeism propensity when parental discrimination is evaluated as unfair, but the relationship diminished with increasing fairness-perceptions. Thus, it can be suggested that observed unfair treatment of other groups may even trigger presenteeism among those who do not belong to the disadvantaged group. However, it must first be considered that at least some participants belonging to the group of childless respondents do not have their own children yet, but may plan to have children in the near future. This can be assumed, for instance, because our sample was restricted to employees in their reproductive ages, and the mean age in the group of childless respondents is considerably smaller than in the group of parents. Hence, although both groups may differ in their actual parental status, those who plan to have children may be equally personally affected by observed parental discrimination. Second, we cannot rule out that observed parental discrimination is simply an indicator of a general latent hostile organizational climate. Hence, like other measures regarding organizational climate are unavailable, we are unable to rule out, for instance, whether the association found is simply an effect of broader organizational distributive injustice, which has been found to be related to presenteeism (Ferreira et al. 2019).

In sum, it can be concluded from multivariate analyses that both observed and experienced parental discrimination is related to an increased presenteeism propensity. This was confirmed for the total sample and, despite some varying results, for both subsamples of childless respondents and parents. In the subsample of participants with children, ob-

served discrimination is (unconditionally) related to presenteeism propensity, and lifetime experienced discrimination is conditionally related to presenteeism propensity in those who disagree with economic justifications of discrimination. In the subsample of childless respondents, lifetime experienced discrimination is (unconditionally) related to presenteeism propensity and observed discrimination is conditionally related to presenteeism propensity in those who perceive discrimination as unfair.

The present study has certain limitations. As the study is based on cross-sectional research, it is not possible to draw conclusions about causality based on our correlational results. However, it seems unlikely that the observed correlational structure stems from a reversed causal relationship (that is, parental discrimination as a consequence of presenteeism behavior) and our results are consistent with previous findings from experimental research (Heiserman and Simpson 2022). The total explained variance in presenteeism propensity was only moderate, which, however, could be expected in the light of previous research that has confirmed a large number of personal and organizational factors affecting presenteeism behavior (Miraglia and Johns 2016).

Although it was possible to rely on validated and frequently used measures for the outcome variables, validated measures regarding experienced and observed parental discrimination as well as for the appraisal of parental discrimination are not available to the best of our knowledge and had to be developed for this study. Our appraisal measures regarding the evaluation of discrimination incidents were based on just two fictional cases, and it remains unclear whether some specific cues embedded in the scenarios presented may have affected respondents' evaluation. Hence, future research could use more elaborate methods, such as vignette designs, to achieve higher external validity of the appraisal measures.

Our measures of sickness absence and sickness presence were based on self-reports. Although the reliability and validity of such self-reports, combined with a large observation period of 12 months, have been questioned, such measures are frequently used in studies on sickness absence and sickness presence (Ruhle et al. 2020). Although sickness presenteeism is only accessible through self-reports, prior research confirmed high agreement (with respect to frequency and correlational structures) between self-reported and register-based data on sickness absence over a 12-month period (Ferrie et al. 2005). Moreover, we have no information regarding the severity of illness that was related to respondents' sickness absence or presence. Hence, it could be argued that presenteeism is predominately related to milder illnesses and that parents will more often attend work despite mild symptoms to keep up with their family responsibilities. However, our measure of sickness presence included a frame of reference (i.e., we asked about days attending work despite sickness that "would have justified taking sick leave") which is recommended for research that is aimed to compare presenteeism and sickness absence (Ruhle et al. 2020). Furthermore, although respondents with children reported a slightly higher number of sickness presence days and a higher presenteeism propensity compared to childless respondents, these differences were not significant. In addition, we have no information on whether respondents were working during the 12-month period. Therefore, individuals who did not work during the entire observation period (for instance those in parental leave) could have reported a smaller number of sickness absence or sickness presence days. However, such bias would not affect presenteeism propensity, which is calculated as the ratio of sickness presence days to the reported total number of sickness absence and presence days.

Moreover, although our sample was drawn from a diverse population with a large variance regarding socioeconomic and professional backgrounds, only a small number of invited people participated in the survey. Hence, self-selection effects must be considered such that individuals with high sensitivity regarding unfair treatment related to parental status are overrepresented in our sample. However, counter to our expectation that predominantly those with children will be motivated to join the survey, approximately half of the participants were childless. Further research employing alternative sample strategies would be necessary to verify the generalizability of our results.

Finally, a sampling strategy with disproportional gender stratification was used because it was expected that females are more frequently affected by parental discrimination. Consequently, the number of male respondents in our sample was too small to allow for deeper analyses separated by respondents' gender. Thus, future research with balanced gender proportions would be necessary for this purpose.

Despite the above-mentioned limitations, our study provides insights into the effects of parental discrimination on sickness presenteeism. Although the amount of explained variance is weak, the results suggest increased presenteeism as employees' reactance behavior to compensate experienced as well as observed discrimination at the workplace. The effect on sickness presence propensity of observed discrimination seems weaker than that of experienced discrimination, and we must consider differences between subsamples. Nevertheless, our results suggest that parental workplace discrimination may not only increase presenteeism of workers with caring responsibilities, but also that of other employees who observe unfair treatment of colleagues. Sickness presenteeism is associated with negative consequences such as strain and impaired health at the individual level (Bergström et al. 2009; Taloyan et al. 2012), but also with adverse effects for organizations, such as increased risks for errors, accidents, and productivity loss (Cooper and Dewe 2008; Niven and Ciborowska 2015; Robertson and Cooper 2011). Our results suggest that parental workplace discrimination may not only increase presenteeism of workers with caring responsibilities, but also of other employees who observe unfair treatment of colleagues. Organizational cultures that build on strong attendance norms and the ideal worker norm will increase such tendencies, whereas implementation of family friendly policies such as options for remote work, flexible time off, or temporary part-time arrangements may help to mitigate presenteeism.

**Supplementary Materials:** The following supporting information can be downloaded at: https://www.mdpi.com/article/10.3390/socsci13010070/s1, Table S1: Robustness-check of bivariate and multivariate results (total sample); Table S2: Robustness-check of bivariate and multivariate results separated for childless respondents and respondents with children.

**Author Contributions:** Conceptualization, J.G. and M.B.-R.; methodology, J.G. and M.B.-R.; formal analysis, J.G.; data curation, J.G.; writing—original draft preparation, J.G. and M.B.-R.; visualization, J.G. All authors have read and agreed to the published version of the manuscript.

**Funding:** Open Access Funding by the University of Linz.

**Institutional Review Board Statement:** Ethical approval was not required for anonymous survey data with informed consent.

**Informed Consent Statement:** Informed consent was obtained from all subjects involved in the study.

**Data Availability Statement:** The data supporting the conclusions of this article can be requested from the authors.

**Acknowledgments:** We thank the Upper Austrian Chamber of Labor for their support with the collection of data.

**Conflicts of Interest:** The authors declare no conflicts of interest.

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
