# Peer review of "Effects of Parental Workplace Discrimination on Sickness Presenteeism"

_socsci, doi:10.3390/socsci13010070_

Round 1
Reviewer 1 Report
Comments and Suggestions for Authors
The perception and experience on discrimination is really important, but you have omitted any reference to the legal regulation.
Does it not seem to you that this is also relevant, as well as the perception of illegitimacy or unfairness? What do workers know about the legal regulation in the country and in the EU?
Author Response
Response to the Reviewers
Many thanks to the reviewer for the valuable comments.
Please find below each of the reviewer’s recommendation and our response to the points raised.
Reviewer #1:
The perception and experience on discrimination is really important, but you have omitted any reference to the legal regulation.
Response:
We added explanations of the legal regulations in Austria.
First, Federal Equal Treatment Act is relevant for the general regulations with respect to different forms of illegal parental discrimination.
This has been added at the beginning of the introduction section:
“Austrian law (Federal Equal Treatment Act, Federal Law Gazette I No. 66/2004 as amended by Federal Law Gazette I No. 107/2013; Wochenalt and McGrew-Taferl, 2020) defines parental workplace discrimination as unequal treatment of employees regarding…”
We have also included notes when Austrian parents have a legal entitlement to reduce working hours and work part-time for a limited period of time to the introduction section:
“Moreover, Austrian mothers and fathers with caring responsibilities for children have a statutory right to request reduced working hours under certain conditions. According to the Maternity Protection Act 1979 employees in companies with more than 20 employees are allowed to reduce their working hours after the birth of their child until the child’s seventh birthday if they have been continuously employed in the same company in the last three years. In doing so, parents have to reduce their working hours by at least 20 percent. The start, duration and extent of the part-time employment have to be arranged with the employer, taking into account the interests of the company as well as the employee’s interests (§ 15h Maternity Protection Act 1979, Federal Law Gazette 221/1979 amended by Federal Law Gazette I 149/2015). Therefore, in many companies denying the request of parents with preschool children to reduce working hours contradicts the legal provisions in Austria and represents a form of parental discrimination”.
Furthermore, as both legal regulations guided our construction of the two fictional case-descriptions, we added the following to the sub-section “moderators”:
“According to Austrian laws women in their reproductive years must not be discriminated based on their prospective caring responsibilities (Equal Treatment Act, Federal Law Gazette I No. 66/2004 as amended by Federal Law Gazette I No. 107/2013). In addition, mothers and fathers with preschool children have a statutory right to request a reduction of their working hours under certain conditions to make it easier for parents to reconcile their work and family duties (§ 15h Maternity Protection Act 1979, Federal Law Gazette 221/1979 amended by Federal Law Gazette I 149/2015). Based on these legal regulations we used two fictional cases to measure respondents’ appraisal of unequal treatment. Although both cases represent an illegal treatment due to parental status according to existing laws, we expect variations of respondents’ perceptions of whether such treatment is fair or acceptable”.
Reviewer #1:
Does it not seem to you that this is also relevant, as well as the perception of illegitimacy or unfairness? What do workers know about the legal regulation in the country and in the EU?
Response:
You are right. It would be interesting to know about employees’ awareness of the legal situation of pregnant women and parents. Previous research analyzed current leave- and other employment-related policies in Europe. But we aren’t aware of any research on subjective perceptions of such legal regulations. Of course, we agree that country specific and comparative research on how employees are informed about their rights would be helpful.
Reviewer 2 Report
Comments and Suggestions for Authors
This is a cross-sectional study of the impact of parental workplace discrimination on presenteeism in the workplace. I found the rationale for the study compelling. The authors did an excellent job of considering competing theoretical models to arrive at their hypotheses. The writing was clear and engaging. The authors demonstrated their command of the current literature and did a brilliant job of explaining the complexities of this research area.
While the rationale for the study and associated hypotheses were strong, I found the measures and reporting of the analysis to be below the expected standard. Perhaps a revision of the manuscript with more explanation about the measures and more detail regarding the analysis would help.
First, some of the questions asked about the past 12 months. Was there any check that the participants were working during the past 12 months? Could some have taken parental leave or perhaps not been employed during the past 12 months? Was this considered. If yes, how was it handled - e.g. were participants to estimate based on the time they had been employed? If not, it is a problem.
It appeared that the measures were developed specifically for this project and had not been used in previous research. Is that correct? If so, what methods were used to develop the questions and how was validity or reliability assured?
The sickness absence and sickness presenteeism measures were based on numbers of days only. There is no measure of severity of illness. It seems that all days of sickness are considered equal, which is a mistake. There is also no indication of whether the participants made different decisions based on their illness being contagious or not. Parental decisions to go to work when there is a mild non-contagious illness may be due to the need to engage with daily activities such as taking the children to school. Parents have to perform a range of tasks when unwell and that may mean they are more likely to go to work compared to childless if they have a mild illness. Because the measure is just number of days, it is difficult to know if the presenteeism applies to milder illness only.
Were the items for the independent variables from previously published research? I was also unsure about calculation of the Cronbach's alpha. Some items were 1-3 and others were 1-4. Please provide more details about the calculation.
I also wondered if the moderators and control variables came from published sources.
If any of the measures came from published sources, please provide details of the reliability, validity or other relevant metrics.
Please provide details of the software used for analysis.
Please provide more details of how you determined that your data met requirements for the bivariate and multivariate analyses you have conducted.
The conclusions are too strong for the data. From Table 2, the R-squared values indicate that there is a lot of variance that hasn't been accounted for in the models. It's not unusual for that to happen with real-world complexities, but it means the interpretation of the results should be more nuanced.
Author Response
Response to the Reviewers
Many thanks to the reviewer for the valuable comments.
Please find below each of the reviewer’s recommendation and our response to the points raised.
Reviewer #2:
First, some of the questions asked about the past 12 months. Was there any check that the participants were working during the past 12 months? Could some have taken parental leave or perhaps not been employed during the past 12 months? Was this considered. If yes, how was it handled - e.g. were participants to estimate based on the time they had been employed? If not, it is a problem.
Response:
Yes, you are right. We can’t rule out that some respondents may have not be employed during the entire observation period of the past 12 months. This may bias the count measures (i.e. number of absence and presence days as well as the sum of both). However, it will not bias presenteeism propensity (which is the primary focus of our analysis). The presenteeism propensity is computed as the number of presenteeism days divided by the sum of presenteeism and absence days (i.e. the proportion of sickness presence days with regard to the sum of days with sickness). Hence, this measure is not substantially affected by different period lengths. We added the following explanation to the limitations in the discussion section:
“Also, we have no information on whether respondents were working during the 12-months period. Therefore, individuals who did not work during the entire observation period (for instance those in parental leave) could have reported a smaller number of sickness absence or sickness presence days. However, such bias would not affect presenteeism propensity which is calculated as the ratio of sickness presence days to the reported total number of sickness absence and presence days.”
Reviewer #2:
It appeared that the measures were developed specifically for this project and had not been used in previous research. Is that correct? If so, what methods were used to develop the questions and how was validity or reliability assured?
Response:
Yes, except for the dependent variables and the single-item measure of job-satisfaction, measures had to be developed specifically for this project. To assure validity, the items for experienced and observed parental discrimination were developed together with two practical experts in the field. This explanation has been added to the measurement-section:
“The items were lent toward different types of parental discrimination as listed in the Austrian Federal Treatment Act and a documentation of frequent cases from an advocacy organization (Wochenalt and McGrew-Taferl, 2020) and were developed by the authors of this paper together with two experts in the field with professional experience in counselling victims of parental discrimination.”
Furthermore, we added the following to the subsection “moderators”:
“While case study research is an increasingly popular approach among qualitative researchers (Creswell, 2013; Priya, 2021; Thomas, 2011), case study approaches in quantitative studies are still rare. But fictional case studies seem to be a suitable approach to evaluate tendencies towards stereotypical judgments, which are methodologically difficult to capture in real settings (Gorman and Mosseri, 2019; Kaiser and Major, 2006; O’Connor and Kmec, 2020). Therefore, we developed our measures on previous qualitative studies (e.g. Byron and Roscigno 2014)”.
Moreover, the following sentence has been added to the limitations in the discussion section:
“Although it was possible to rely on validated and frequently used measures for the outcome variables, validated measures regarding experienced and observed parental discrimination as well as for the appraisal of parental discrimination are not available to the best of our knowledge and had to be developed for this study.”
Reviewer #2:
The sickness absence and sickness presenteeism measures were based on numbers of days only. There is no measure of severity of illness. It seems that all days of sickness are considered equal, which is a mistake. There is also no indication of whether the participants made different decisions based on their illness being contagious or not. Parental decisions to go to work when there is a mild non-contagious illness may be due to the need to engage with daily activities such as taking the children to school. Parents have to perform a range of tasks when unwell and that may mean they are more likely to go to work compared to childless if they have a mild illness. Because the measure is just number of days, it is difficult to know if the presenteeism applies to milder illness only.
Response:
Whether disease severity should be considered in presenteeism measures is contested (see discussions on that issue in the position paper of Ruhle et al. 2020). Halbesleben et al. 2014 and Johns 2010 for instance, argue that the probability to decide for sickness presence instead of sick leave will decrease with illness severity for logical reasons (although the empirical proof of that assumption is still missing to our knowledge). However, following the recommendations of Ruhle et al. (2020) we decided to use a measure that includes a “frame of reference” instead of “context free” measures. Instead of asking how many days employees attended work despite sickness we asked about attending work despite sickness that “would have justified taking sick leave”. Such measures are intended to rule out that respondents also think of minor discomfort in reporting presenteeism and are recommended for research that is aimed to compare sickness absence and sickness presence behavior. However, we have to note that it was not the aim of our research, to explicitly analyze health risks for the respondents or their colleagues induced by presenteeism behavior. Therefore, we did not include questions regarding the types of illness (e.g. contagious or non-contagious) related to presenteeism. The supposition that parents may attend work with mild diseases more often due to their caring responsibilities seems plausible. However, we found no significant differences in the number of presenteeism days and presenteeism propensity between childless respondents and those with children, which is applicable from Table 1. Moreover, our results suggest that the association between discrimination and presenteeism seem to hold for both groups separately.
We added the following explanation to the limitations in the discussion section:
“Moreover, we have no information regarding the severity of illness that was related to respondents’ sickness absence or presence. Hence, it could be argued that presenteeism is predominately related to milder illnesses and that parents will more often attend work despite mild symptoms to keep up with their family responsibilities. However, our measure of sickness presence included a frame of reference (i.e. we asked about days attending work despite sickness that “would have justified taking sick leave”) which is recommended for research that is aimed to compare presenteeism and sickness absence (Ruhle et al., 2020). Furthermore, although respondents with children reported a slightly higher number of sickness presence days and a higher presenteeism propensity compared to childless respondents, these differences were not significant.”
Reviewer #2:
Were the items for the independent variables from previously published research? I was also unsure about calculation of the Cronbach's alpha. Some items were 1-3 and others were 1-4. Please provide more details about the calculation.
Response:
Please also see our response regarding the development of measures for the independent variables above.
With respect to Job satisfaction (which was measured with a single item that has been used in many studies) we added a reference (Quinn et al. 1974) and a second reference that indicated an acceptable measurement quality of single-item measures for job satisfaction (Wanous et al. 1997)
With respect to the different scale range (1-3 and 1-4) of items concerning experienced discrimination, we added more details about calculations. Different response categories were necessary due to logical reasons. We transformed all items to the same scale range (i.e. continuous values between 0 and 1) before computing the mean score (and Cronbach’s Alpha). In our first version of the paper we did not include Cronbach’s Alpha for this measure, as we think that items regarding the experience of different types of discrimination incidents constitute a type of formative and not a reflective measure. However, for consistency reasons, we now added Cronbach’s Alpha.
The measurement description has been updated in the following way:
“For the computation of an overall index of experienced discrimination, all answer categories were transformed to continuous values between 0 (representing no experience) and 1 (highest intensity of experienced discrimination) (i.e. 3-point scales: original values 1, 2, 3 were transformed to 0, 0.5, 1; 4-point scales: original values 1, 2, 3, 4 were transformed to 0, 0.33, 0.66, 1). Subsequently, the index was computed as the mean score across all items (Cronbach’s Alpha=0.93). “
Reviewer #2:
If any of the measures came from published sources, please provide details of the reliability, validity or other relevant metrics.
Response:
Please see our response regarding the development of measures for the independent variables above.
Reviewer #2:
Please provide details of the software used for analysis.
Response:
We now include details of the software used in the analyses-section.
Reviewer #2:
Please provide more details of how you determined that your data met requirements for the bivariate and multivariate analyses you have conducted.
Response:
Although OLS-regression has been proven to be robust against violation of distributional violations, we applied robustness-checks of our calculations. We replicated the bivariate analyses concerning Pearson’ r with the calculation of Spearman’s rank-correlation. The multivariate regressions were replicated with help of generalized linear models. For this purpose, we applied zero-inflated negative binomial models to the count measures (sickness presence days, sickness absence days and days with sickness) and regression models with a binomial probability distribution and a logit link function for presenteeism propensity. The results of these robustness checks are provided as supplementary material. As these analyses showed equivalent results, we continued to use the results of the OLS regressions due to better interpretation.
In addition to the supplementary files, we added the following explanation to the analyses-section:
“Because the distributional requirements to compute Pearson’s correlation and OLS-regressions may be violated, robustness checks by using alternative model-estimations were conducted. With respect to the count measures (days with sickness, sickness absence, sickness presence), the analyses were replicated with zero-inflated negative binomial regression models (Bierla et al., 2013; Gerich, 2016). Presenteeism propensity is a continuous proportional measure including one- and zero cases. Therefore, a generalized linear model with a binomial probability distribution and a logit link function was used for the robustness-check regarding for this outcome (Papke and Wooldridge, 1996). All bivariate correlations were replicated by Spearman’s Rho coefficient. The results of these robustness-checks are available from the supplemental material. As the results of the robustness-checks revealed no substantial differences compared to the OLS regressions, we use the results of the latter due to better interpretation. The main analyses were performed with SPSS 29 (IBM Corp., 2022), the robustness-checks were performed with Stata SE 14 (Stata Corp., 2015)”.
Reviewer #2:
The conclusions are too strong for the data. From Table 2, the R-squared values indicate that there is a lot of variance that hasn't been accounted for in the models. It's not unusual for that to happen with real-world complexities, but it means the interpretation of the results should be more nuanced.
Response:
We think that the R-square value is comparable to other (isolated) factors that were found to be associated with presenteeism. However, we added the following statement to the limitations in the discussion section:
“The total explained variance in presenteeism propensity was only moderate, which however, could be expected in the light of previous research that has confirmed a large number of personal and organizational factors affecting presenteeism behavior (Miraglia and Johns, 2016). “